# Titanium-45 (^45^Ti) Radiochemistry and Applications in Molecular Imaging

**DOI:** 10.3390/ph17040479

**Published:** 2024-04-09

**Authors:** Shefali Saini, Suzanne E. Lapi

**Affiliations:** 1Department of Radiology, University of Alabama at Birmingham, Birmingham, AL 35294, USA; sfnu@uab.edu; 2Department of Chemistry, University of Alabama at Birmingham, Birmingham, AL 35294, USA

**Keywords:** titanium-45, PET, molecular imaging, radiopharmaceutical

## Abstract

Molecular imaging is an important part of modern medicine which enables the non-invasive identification and characterization of diseases. With the advancement of radiochemistry and scanner technology, nuclear medicine is providing insight into efficient treatment options for individual patients. Titanium-45 (^45^Ti) is a lesser-explored radionuclide that is garnering increasing interest for the development of positron emission tomography (PET) radiopharmaceuticals. This review discusses aspects of this radionuclide including production, purification, radiochemistry development, and molecular imaging studies.

## 1. Introduction

Titanium is the ninth most abundant metal in the earth’s crust and the second most abundant transition metal [1]. Titanium has been widely incorporated in many applications over the last few decades including personal hygiene products (including shampoo, deodorant, toothpaste, and sunscreens), medical implants, coating materials (in the paint industry), and food products (including dressings and beverages) [2,3]. 

Titanium metal has a very low solubility in water while the aqueous Ti(IV) has a high propensity for hydrolysis [4]. One of the main reasons why the radiochemistry of Ti has been underdeveloped is because a stabilizing ligand is always required with titanium to minimize hydrolysis. Titanium has been used in several preclinical studies as a component of anticancer drugs for several tumor types [5]. Titanocene dichloride was one of the first titanium-based drugs to be used as an antitumor drug and proceeded to phase two of clinical trials. Though this compound was designed to be a stabilized complex through the incorporation of two cyclopentadienide ligands and two chloride molecules, titanocene dichloride was not further investigated due to its suboptimal stability [6]. Though several modifications have been made to the structure of Ti compounds in efforts to create more suitable complexes [7,8,9,10,11,12,13], there are no anticancer Ti compounds currently under investigation in clinical trials reported on clinicaltrials.gov (https://clinicaltrials.gov/search?cond=Cancer&intr=titanium (accessed on 26 December 2023); database screened using search terms “cancer”, “titanium”). 

On a radiotracer scale, applications of titanium have been investigated using two radiotitanium isotopes: titanium-44 (^44^Ti) and titanium-45 (^45^Ti). Titanium-44 is a radioisotope with a very long half-life of 60 years and potential applications as part of a ^44^Ti/^44^Sc generator system [14,15,16]. The decay of ^44^Ti to ^44^Sc (t_1/2_ = 3.97 h, mean E_β+_ = 0.632 MeV (branching ratio (BR) 94.27%)), and the secular equilibrium due to the half-life difference makes this a potential source for a constant supply of ^44^Sc at research sites [16,17,18]. 

Titanium-45 is of interest for positron emission tomography (PET) applications due to the characteristic properties of this radionuclide including its half-life of 3.08 h, its decay by positron emission (BR 85%), its low mean positron energy (E_β_^+^ 439 keV) and very small fractions of other gamma rays (720 keV (0.15%)) [19,20]. The half-life of 3 h is also ideal for imaging small molecules. Table 1 summarizes the properties of titanium-45 compared to the other commonly used radionuclides in PET imaging. Titanium-45 decay exhibits a similar positron energy to ^11^C (386 keV) [21] and significantly less than that of ^68^Ga (830 keV) [22]. Since the mean positron energy is inversely proportional to the maximum spatial resolution obtainable with PET, ^45^Ti may enable better image resolution than ^68^Ga in imaging studies with small animal PET scanners.

This review focuses on the status and recent progress of ^45^Ti radiopharmaceutical development including production, purification, radiochemistry development and molecular imaging studies. 

## 2. Production and Purification

The production route of a radionuclide and the cost of production are important considerations for the generation of quantities suitable for clinical applications. The production of ^45^Ti has been achieved through the proton irradiation of ^nat^Sc target material using a (p,n) reaction route (Figure 1A) [23,24,30,31,32,33,34,35,36,37,38,39]. In addition to the high cross-section for this reaction (Figure 1B) [40], another advantage of ^45^Ti production is the monoisotopic target material (^45^Sc). Since the ^45^Sc target is available in 100% natural abundance, the target material of a high purity is readily commercially available. This also eliminates the need for investment in an enriched material, making production cost-efficient. The reported titanium-45 production methods are provided in Table 2. 

### Production of Titanium-45

Titanium-45 production was investigated by Nelson et al. in 1964 using a (p,n) reaction from ^nat^Sc targets [41]. Irradiated targets were dissolved in 6 M of HCl and ^45^Ti in the dissolved fraction was oxidized from +3 to +4 by the addition of a small amount of concentrated nitric acid (HNO_3_). After drying and reconstitution for three cycles, the solution was loaded onto an AG 50W x 8 column (100–200 mesh) and eluted in 6 M of HCl. These fractions were dried and reconstituted in 1 M of HCl as a probable [^45^Ti]TiOCl_2_ complex. This separation procedure took 2 h with 75–90% recovery yields [41]. Later reports for the production of ^45^Ti by Merill et al. [20] reported a ^45^Ti separation using Dowex 1-X8 resin with elution in 8 M of HCl with an overall radiochemical yield of 30%. Electroplated Sc targets were also examined, and a 99.8% radionuclidic purity was achieved at the end of the bombardment [20]. 

Given the short half-life of titanium-45, separation methods with short processing times are ideal. Therefore, Vāvere et al. modified the previous procedure by Nelson et al. for ^45^Ti production with a focus toward more rapid purification [23]. This purification method used a target dissolution in 6 M of HCl and purification using AG 50W-X8 resin followed by the evaporation of the ^45^Ti elution to dryness. The dried product was reconstituted in an appropriate solvent to the desired volumes. An average recovery of 92.3% was obtained using this separation method without the need for HNO_3_ or several drying steps. 

To ensure suitable stability under a higher beam current and for the development of robust targets, sedimentation techniques have also been tested for ^45^Ti target preparation. Sadeghi et al. investigated the deposition of a Sc_2_O_3_ target material on copper backings [42]. The Sc_2_O_3_ solution was slowly evaporated over 12–48 h at room temperature. Several solvent systems including ethyl cellulose (EC) and methyl cellulose (MC) were investigated for better target adhesion and stability and the results were confirmed using scanning electrode microscopy (SEM). A 25–30% MC in the Sc_2_O_3_ solution was found to provide the best adhesion. 

Gagnon et al. [32] and Price et al. [45] investigated the purification of ^45^Ti using a hydroxamate-based resin. This hydroxamate resin has been previously used for ^89^Zr separation by the modification of a carboxymethyl (CM) weak cation exchange resin, as described by Holland et al. [46]. Since both Ti and Zr belong to the same group in the periodic table and have similar chemistry and oxidation states, it was hypothesized that hydroxamate resin could potentially be a suitable resin for the separation of ^45^Ti from a Sc target material. Gagnon et al. compared the results for unreacted carboxymethyl resin and the hydroxamate functionalized resin and found that while the loading efficiency of the CM resin was <5%, the hydroxamate resin resulted in an ~74% loading efficiency using 2 M of HCl. The elution of the ^45^Ti was performed in 1 M of oxalic acid in 3 × 200 µL fractions for the unreacted CM resin and 4 × 300 µL for the hydroxamate resin. Overall, a 56% recovery yield was observed for hydroxamate resin with 32% of the total activity obtained being the highest amount in a single fraction (300 µL). Similarly, Price et al. reported an elution in 1 M of oxalic acid (5 × 1 mL). However, the chemical separation resulted in a significant breakthrough of ^45^Ti in the washes and 30% activity in the final elution fraction [45]. Chaple et al. further optimized the separation by investigating different elution methods for ^45^Ti using hydroxamate resin [24]. The loading and wash conditions were kept the same, but the elution method investigated the use of citric acid instead of oxalic acid. This separation method reported a 78 ± 8% recovery of ^45^Ti in 3 mL of citric acid (1 M) [24].

Pederson et al. investigated a liquid–liquid extraction (LLE) technique [34]. Additionally, to overcome the radiation exposure from manual processing, the investigators developed an automated procedure for the separation of radioactive material using a membrane phase separation.^45^Ti was separated in a solution containing a mixture of guaiacol/anisole (9:1). The extraction of ^45^Ti was 84.8% recovered from 0.01 M of solution as TiCl_4_.

Giesen et al. further investigated a “one pot” method of ^45^Ti purification using a solvent-free approach (Figure 2). This method involved a thermochromatographic separation employing dry distillation [35]. Briefly, the irradiated Sc foil was placed in a separation chamber at 900 °C, which converted the ^45^Ti to [^45^Ti]TiCl_4_ by flushing chlorine gas at a flow rate of 15 mL/min. The difference between the boiling point of ScCl_3_ (b.p. 975 °C) and TiCl_4_ (b.p. 136 °C) resulted in the sublimation of [^45^Ti]TiCl_4_, which was collected in a receiving flask [35]. Maximum recovery yields of 53% were obtained at a gas flow rate of 150 mL/min with a 900 °C reaction temperature and 12% Cl_2_ concentration. 

A deuteron-induced reaction pathway was also investigated for ^45^Ti production. Tsoodol et al. investigated the ^45^Sc(d,2n)^45^Ti production route up to 24 MeV of incident beam energy [47]. The production of ^45^Ti using the deuteron reaction route was also investigated previously by Hermanne et al. [48]. The cross-section data showed ^45^Ti could be produced at 15 MeV without a ^44^Ti contamination (t_1/2_ = 60 y) with a high cross-section (σ range = 250–350 mb) [40]. This method did not report any further purification methods. 

Another study investigating several approaches for the separation of ^45^Ti from ^45^Sc using ion exchange chromatography was reported by Strecker et al. [44]. This study investigated the impact on yields by varying the pH of the eluting solvent, adding/removing wash steps and varying the resin mass for several solvent systems. A mixture of MeCN/H_2_O_2_ (0.65 M) resulted in a high recovery yield of 82%. However, with this method, the authors suspected the partial elution of the hydroxamate functional group from the resin. Oxalic acid (0.1 M, pH 2.8) being used as the eluent enabled a 61 ± 8% recovery of ^45^Ti in under 10 min [44]. 

As discussed above, the relatively straightforward production of ^45^Ti puts it at an advantage compared to some other radionuclides such as ^43^Sc (t_1/2_ = 3.89 h, E(β^+^_avg_) = 476 keV, BR = 88%) and ^44^Sc (t_1/2_ = 3.97 h, E(β^+^_avg_) = 632 keV, BR = 94%). Despite their comparable half-lives and decay properties, their limited production capabilities can limit the availability of these radionuclides for clinical applications. Though a ^44^Ti/^44^Sc generator would be ideal to overcome this problem, there have been significant efforts to find suitable loading and elution conditions without the breakthrough of ^44^Ti [14,17,18,39]. 

## 3. Imaging Applications and Chelation Chemistry of Titanium-45

Titanium-45 has a half-life of 3.08 h which is sufficient for the PET imaging of a range of targeting conjugates with short biological half-lives [20]. The mean positron energy of ^45^Ti (439 keV) is almost half that of the well-optimized and routinely used ^68^Ga (830 keV) leading to high resolution images on small animal PET scanners [36]. Additionally, ^45^Ti has an almost 3-fold longer half-life than ^68^Ga (68 min), thus potentially enabling longer time-point scans. Figure 3 shows the structures of the radiolabeled ^45^Ti-complexes discussed in this review.

Vavere et el. conducted the first imaging study with titanium-45 using phantoms to evaluate the image resolution using a small PET scanner. Phantom studies yielded images with comparable resolution to ^18^F (t_1/2_ = 110 min, E_av_(β^+^) = 250 keV, BR = 97%), a clinically approved radionuclide with well-optimized chemistry [23]. 

The radiopharmaceutical potential of ^45^Ti (eluted as [^45^Ti]TiOCl_2_) (Figure 3A) with radiolabeled Human Serum Albumin (HSA), diethylenetriaminepentaacetic acid (DTPA) (Figure 3B) and citric acid (CA) (Figure 3C) was investigated by Ishiwita et al. [49]. Radiolabeling studies were performed at pHs of 4–6 and results were confirmed with paper electrophoresis and TLC. Tissue biodistribution studies were analyzed in rats bearing AH109A hepatomas injected with [^45^Ti]TiO-DTPA and [^45^Ti]TiO-CA [49]. The tissue biodistribution was similar with all three injected compounds with a high blood activity and retention even after 6 h post injection. The uptake in the tumor regions for [^45^Ti]TiO-DTPA and [^45^Ti]TiO-CA was half of the blood pool activity at 6 h post injection. Biodistribution studies were reported for [^45^Ti]Ti-HSA in normal rats and the plasma transferrin binding of ^45^Ti was confirmed as a potential measure for circulating plasma volume measurement. The affinity of Ti with transferrin was also determined using a [^45^Ti]TiO-DTPA complex where DTPA was displaced by transferrin, as confirmed by TLC analysis. In vitro plasma binding studies with [^45^Ti]Ti-HSA were not reported potentially due to the low recovery (30%) of this complex by gel filtration chromatography. The utility of [^45^Ti]TiO-DTPA was also assessed as an indicator of blood–brain barrier (BBB) disruption. Damaged BBB was identified in rats injected with [^45^Ti]TiO-DTPA using the autoradiography of rat brain slices where the uptake was 9x higher in damaged regions than in normal brain regions [49]. A comparison study was also performed by Miura et al. to utilize the potential of [^11^C]C-CoQ_10_ for myocardial imaging, [^18^F]FDM to evaluate the glucose metabolism and [^45^Ti]Ti-DTPA to analyze the blood volume in a dog cardiac model [50]. Initially, dogs were injected with 350 µCi of [^11^C]C-CoQ_10_ and myocardial images were generated every 15 min. Following the [^11^C]C-CoQ_10_ scan, 950 µCi of [^45^Ti]Ti-DTPA were injected in the same animal after 2 h and image acquisition was performed in 12 × 5 min frames. Finally, 3.5 h post the injection of [^45^Ti]Ti-DTPA, 3.58 mCi of [^18^F]FDM were injected, and images were acquired in 8 × 5 min frames. Arterial blood collection was also performed during every study. The [^45^Ti]Ti-DTPA scan was used to correct for the blood spillover between the scans to determine the differential absorption ratio. The imaging data indicated that while [^11^C]C-CoQ_10_ alone could not differentiate between the myocardium and blood pool with a high accumulation in the blood, [^18^F]FDM indicated a high absolute uptake in the myocardium with the [^45^Ti]Ti-DTPA correction [50]. 

Pederson et al. investigated radiolabeling using ^45^Ti purified with an LLE separation with a previously synthesized compound (salan)Ti(dipic) (Figure 3E) [43]. The complex was radiolabeled within 15 min as confirmed by HPLC and radio-TLC but the stability of this complex was not reported [34]. Recently, a PSMA conjugate of a previously optimized functionalized salan-dipic compound was developed to investigate the stability of salan complexes in vivo [36]. Structure modifications to a PSMA-1007 moiety were developed using molecular docking and a linker containing chelidamic acid was used to create salan-Ti-CA-PSMA. The structural modifications were performed with an aim to increase the hydrophobicity of the ligand which can further reduce the kidneys’ uptake. Radiolabeling optimization studies showed a 47% yield with 15 mM of salan and CA-PSMA at 1 h of incubation at 60 °C, which increased to 80–85% when heated at 80 °C. Due to the large reaction volume (6 mL), fractions were collected using preparative HPLC and the elution was concentrated in a PBS/ethanol (9/1) mixture using a C18 cartridge which further reduced the final radiochemical yield to 5.1 ± 2.3% (non-decay corrected). In vitro stability in PBS and mouse serum indicated signs of decomplexation with 33% and 80% intact complexes at 4 h post incubation, respectively [36]. In vivo imaging studies in mice with PC3 tumors indicated an uptake in the gallbladder and intestine while a low uptake was observed in the tumor (1.1% ID/g). The compound was excreted via the hepatobiliary route which was apparent in the PET images (Figure 4). The authors anticipated the instability of this complex due to the decomplexation of [^45^Ti]-salan-Ti-CA-PSMA to [^45^Ti]-salan-Ti-citrate and the further formation of [^45^Ti]Ti(citrate)_2_]^2−^. An improvement of this ligand design was proposed to further improve the stability of this complex. 

Following a citric acid elution purification route, Chaple et al. investigated ^45^Ti radiochemistry with a deferoxamine (DFO) chelator (Figure 3F) [24]. The reactions were incubated at 37 °C in a HEPES buffer (1 M) at a pH of 10. The radiolabeling of DFO with ^45^Ti was reported with molar activities of 1.8 MBq/nmol (48 µCi/nmol). The complex was significantly intact at the 3 h (91 ± 5%) and 6 h (85 ± 11%) incubation time points in the mouse serum [24]. Another recent study investigated the PSMA targeting vector using DUPA conjugated with two different chelating groups: linear desferrichrome (LDFC) and deferoxamine (DFO) including radiochemistry, in vitro cell studies and in vivo imaging studies in tumor xenograft models [37]. Radiochemistry studies of LDFC-DUPA indicated an efficient binding at a pH of 9 at 50 °C in 30 min while DFO-DUPA required a pH of 11 at 37 °C at 1 h of incubation to achieve > 95% radiolabeling. In vitro serum stability studies of [^45^Ti]Ti-LDFC-DUPA showed significant decomplexation at 1 h post incubation with a 33 ± 11% intact complex while [^45^Ti]Ti-DFO-DUPA was 86 ± 9% intact after 6 h of incubation. In vitro cell studies showed a significant binding of [^45^Ti]Ti-DFO-DUPA with the PSMA + LNCaP cell line (0.6 ± 0.1%) compared to the PSMA—PC3 cell line (0.1 ± 0.01%) at 1 h of incubation [37]. Tumor xenograft models were used for PET/CT imaging and showed a tumor uptake of 2.3 ± 0.3% ID/g for LNCaP tumor-bearing mice compared to 0.1 ± 0.1% ID/g for PC3 tumor-bearing mice. However, the complex indicated significant signs of decomplexation as indicated by the presence of radioactivity in off-target organs such as the heart (~10% ID/g), blood (2.5% ID/g), lungs (~12% ID/g), and liver (3.5% ID/g), as also observed with unchelated ^45^Ti. 

Hydroxypyridinone and catechol functional groups have been investigated for the chelation of ^45^Ti. Thermodynamic speciation studies were explored for a wide range of pH values and both catechol and deferiprone ligands were promising with a 1:3 complex formed at a physiological pH [38]. Recently, hexadentate chelators THP^Me^ (Figure 3G) and TREN-CAM (Figure 3H) were investigated for labeling efficiency with ^45^Ti [38]. Radiolabeling yields of 0.28 mCi/nmol (10.36 MBq/nmol) and 0.07 mCi/nmol (2.59 MBq/nmol) were obtained for TREN-CAM and THP^Me^, respectively. Both complexes were 100% intact in a mouse serum for an incubation time of 6 h. The biodistribution of [^45^Ti]Ti-THP indicated signs of decomplexation as evidenced by radioactivity in the blood (1.89 ± 0.12% ID/g) and heart (3.3 ± 0.4% ID/g). [^45^Ti]Ti-TREN-CAM had a significantly different biodistribution pattern and the complex was excreted through the renal pathway while a negligible uptake was observed in the blood (0.05 ± 0.01% ID/g) and heart (0.16 ± 0.01% ID/g) in support of a kinetically inert complex. In a follow-up study by Koller et al., a comparison of the labeling efficiency was performed with a citrate elution and a chloride elution with the TREN-CAM ligand. A nearly 7-fold increase in apparent molar activity was observed with Ti-TREN-CAM for the chloride (6.2 MBq/nmol) elution versus the citric acid (0.74 MBq/nmol) [39]. 

Building on the prior work with a [^45^Ti]Ti-THP chelator, a study with the PSMA-targeting vector was reported by Saini et al. [51] This THP-PSMA-targeting conjugate is in ongoing clinical trials with ^68^Ga for imaging applications [52]. Radiolabeling studies were conducted with [^45^Ti]Ti-THP-PSMA to investigate its labeling efficiency under mild labeling conditions. Contrary to [^45^Ti]Ti-THP^Me^, in vitro serum stability studies of the [^45^Ti]Ti-THP-PSMA complex showed signs of instability with as 82.6 ± 7.3% intact complex at 1 h and 63.6 ± 8.5% intact at 3 h post incubation in a mouse serum. In vitro cell studies illustrated significant cell uptake in the PSMA-expressing cell line (LNCaP) (11.9 ± 1.5% bound) compared to the PSMA non-expressing cell line (PC3) (1.9 ± 0.4% bound). This complex was further investigated in tumor xenograft studies with LNCaP (PSMA +) and PC3 (PSMA –) cell lines. While there was a significantly higher uptake in LNCaP cells (1.6 ± 0.3% ID/g) as compared to PC3 cells (0.4 ± 0.2% ID/g), a low overall tumor uptake was observed in LNCaP tumors. Control studies were performed with [^68^Ga]Ga-THP-PSMA, and though the tumor uptake was not significantly different between ^68^Ga (1.79 ± 0.54% ID/g) and ^45^Ti (1.67 ± 0.26% ID/g) at 2 h post-injection, the overall biodistribution was significantly different for [^68^Ga]sGa-THP-PSMA, with a low background in PET/CT images and no signs of decomplexation with [^68^Ga]Ga-THP-PSMA (Figure 5) [51].

Porphyrin systems have been widely used in metal chelation and recently were investigated for radiolabeling with ^45^Ti by Yekany et al. [53]. A tetrakis(pentafluorophenyl) porphyrin (TFPP) (Figure 3I) complex was synthesized and radiolabeling reactions were carried out with 10 mCi of ^45^Ti and 409 nmol of a TFPP stock in an acetate buffer. The radiolabeling solution required a one-hour reflux and analysis was performed using TLC. Biodistribution results were promising with a low uptake in the blood, heart, skin, and bones at 1 h and 2 h post-injection [53]. Overall, more work with porphyrins coupled with targeting conjugates is required to understand the full potential of porphyrin systems for ^45^Ti complexes. 

More recently, hexadentate 1,2-hyrdoxypyridinone complexes were also analyzed by Carbo-Bague et al. A tripodal chelator HOPO-O_6_-C4 (Figure 3J) was synthesized and radiolabeling efficiency was investigated with ^45^Ti for PET imaging [54]. The radiolabeled complex was synthesized and analyzed using HPLC with radiolabeling yields of 0.06 mCi/nmol (2.2 MBq/nmol) obtained at mild labeling conditions of pH 7 at 37 °C [54]. A similar trend was observed in previous studies with the 3,4-hydroxypyridinone complex described by Koller et al. [38]. Crystal structures identified the formation of both *fac-* and *mer*-isomers for Ti^4+^ yielding two peaks in HPLC analysis. While stability studies in the presence of a mouse serum showed an 81 ± 1.6% intact complex after 6 h of incubation, in vivo biodistribution analysis of this complex at 3 h post injection showed a likely significant decomplexation with uptake in the heart, lungs, and bones [54]. 

Vavere et al. reported a detailed study involving radiolabeling and in vivo imaging analysis with [^45^Ti]Ti-transferrin [31]. The radiolabeling of [^45^Ti]Ti-transferrin was performed by incubating 37 MBq of ^45^Ti with 10 µg of apotransferrin in a HEPES-buffered saline solution. The incubation of this reaction mixture was carried out for 5–60 min at room temperature and the Ti-transferrin solution was eluted for further studies using a PD-10 column. In vivo studies were performed via the co-injection of [^45^Ti]Ti-transferrin with [^67^Ga]Ga-citrate as a control due to its known affinity towards transferrin. Tumor models were developed by injecting EMT-6 murine mammary carcinoma cells in female mice. [^45^Ti]Ti-transferrin and [^67^Ga]Ga-citrate were injected and biodistribution studies were performed at 2 h, 4 h, and 24 h post-injection. While there was an observable uptake in tumors for [^45^Ti]Ti-transferrin, the SUVs in tumors were modest (SUV_mean_ ~ 0.5) compared to those of other non-target organs (muscle SUV_mean_ 0.3). Tissue biodistribution studies indicated similar biodistribution profiles between [^45^Ti]Ti-citrate and [^45^Ti]Ti-transferrin potentially due to transchelation. Additionally, the biodistribution of TiOCl_2_ was significantly different than that of [^45^Ti]Ti-transferrin, supporting the hypothesis that [^45^Ti]Ti-citrate is more appropriate for enabling a binding to transferrin [55]. 

As discussed, reports in the literature indicate that titanium rapidly binds with the iron-binding protein transferrin [56]. In biological systems, the binding ratio between iron (Fe) and transferrin is 1:3000, indicating that there are potentially many metal-binding sites available for other metals. In addition to Fe^3+^, transferrin has been shown to interact with Aluminum (Al^3+^), Vanadium (V^4+^), Manganese (Mn^2+^), Ruthenium (Ru^3+^), Gallium (Ga^3+^) and Titanium (Ti^4+^) [57,58,59]. Several reports suggest that the binding potential of Ti^4+^ is higher than that of iron for transferrin [4,56]. Additionally, since the tumor cell surface has elevated levels of transferrin receptors, transferrin could also potentially be used for tumor imaging [31]. 

Anticancer drugs like titanocene dichloride were the first non-platinum metal-based drugs to enter clinical trials [60]. However, solubility challenges have affected the further development of this class of compounds for human use [61]. To address solubility challenges, Severin et al. explored the salan ligand to improve the radiochemistry of ^45^Ti and improve the solubility of the resulting complexes [43]. A diol-functionalized resin was synthesized and the on-column synthesis of [^45^Ti](salan)Ti(dipic) was investigated with ^45^Ti using an automated module. The elution from the diol resin was loaded on a Sep-Pak C18 cartridge to elute the radiolabeled complex in ethanol. Overall yields were ~30% from the first column while the final elution yields were 15% from the second cartridge [43]. An in vivo investigation of the resin-synthesized [^45^Ti](salan)Ti(dipic) was carried out with mice-bearing HT-29 colorectal adenocarcinoma tumors. The tumor uptake was ~1% ID/g while the in vivo stability data showed no decomplexation of the compound with hepatic clearance. Differences between the (no carrier added intravenous) *ncaiv* and (no carrier added intraperitoneal) *ncaip* administration routes were also utilized to investigate the differences in biodistribution. The *ncaip* administration route showed a high accumulation in the intestine and a 0.5% tumor uptake while the *ncaiv* administration route showed an accumulation in the liver and intestine with ~0.4% tumor accumulation. The complex was observed to have a high lipophilicity and the authors suggest a structural change to increase the in vivo circulation time, which could potentially increase in vivo tumor uptake.

Considering the challenges associated with titanium chemistry due to rapid hydrolysis under aqueous conditions, alternative complex formation methods have also been investigated. A chelator-free approach using mesoporous silica-based nanoparticles was investigated by Chen et al. for ^45^Ti [33]. The in vivo evaluation of the synthesized nanoparticles was performed in 4T1 tumor (breast cancer model)-bearing mice [33]. However, the complex was less stable in the presence of a serum protein (85% intact at 3 h of incubation) than in previous ^89^Zr-based studies (>96% intact). The affinity for ^45^Ti was also investigated with -NH_2_-modified nanoparticles (MSN) [62]. [^45^Ti]TiMSN-PEG_5K_ was synthesized via the incubation of MSN-NH_2_ (500 µg) with ^45^Ti (0.5 mCi) at a pH of 7 for one hour at 75 °C with a 90% yield. After several washings with PBS to remove the excess PEG, the synthesized [^45^Ti]Ti-MSN-PEG was used for in vivo imaging studies. Mice were imaged at 0.5 h, 5 h and 14 h with free [^45^Ti]Ti-oxalate as a control group. Mice injected with [^45^Ti]Ti-oxalate had a higher uptake in the tumor than that of [^45^Ti]Ti-MSN likely due to the affinity of Ti with transferrin (Figure 3D) which may help regulate the delivery of Ti to the tumor site. A high blood uptake and high tumor uptake was observed with mice injected with [^45^Ti]Ti-oxalate (3.0% at 30 min and 14.3% at 14 h) [33]. In the MSN imaging group, the tumor uptake was significantly less than that of [^45^Ti]Ti-oxalate (1.7% ID/g at 3 h and 4.6% ID/g at 6 h). Intrinsic radiolabeling using nanoparticles is a novel approach; however, to improve in vivo targeting, modifications to the silica-based nanoparticle surface are required for successful imaging applications.

## 4. Conclusions

In an exciting era of nuclear medicine, research on novel radionuclides for imaging and therapeutic applications has been mounting to develop agents for personalized patient care for different diseases. Titanium-45 is a radionuclide with properties suitable for PET imaging radiopharmaceuticals. Its production and purification have been widely explored in the last several decades, utilizing ion exchange chromatography [23,32,42], solvent-free methods [35] as well as resin chelation [43]. Several chelating moieties including salan, hydroxamate, hydroxypyridinones, catechol, and deferiprone have been investigated for potential imaging applications in nuclear medicine with ^45^Ti. Deferiprone, porphyrin and catechol have the highest stability in vitro while the catechol and porphyrin complexes were most promising among all the chelators with high in vivo stability. Studies in the literature investigating the radiopharmaceutical potential of ^45^Ti have been reported for various disease models such as prostate cancer, colorectal cancer, and breast cancer. However, its low in vivo stability has been a limiting factor in determining the full potential of this radionuclide for PET imaging applications. The challenges associated with coordination chemistry and ligand design provide an opportunity to explore the strategies that are unique to the characteristic properties of titanium. Thus, it is important to design ligand systems that can form stable complexes with ^45^Ti and can be functionalized without affecting the stability of the complex. Further studies are warranted with a focus on the structural modification of existing ligands to improve the stability for the translation of these ligands into radiopharmaceuticals. Though the search for highly stable chelators for ^45^Ti is still ongoing, encouraging results point to a path forward for the development of ^45^Ti radiopharmaceuticals. 

## Figures and Tables

**Figure 1 pharmaceuticals-17-00479-f001:**
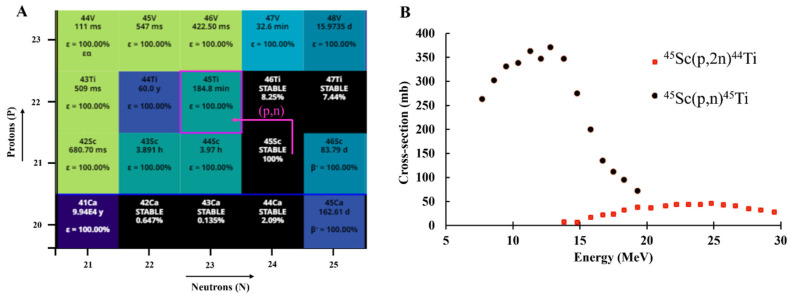
(**A**) Chart of the nuclides section illustrating the nuclear reaction route for ^45^Ti production [30] and the (**B**) Cross-sections for proton-induced reactions for ^45^Ti and ^44^Ti production [40].

**Figure 2 pharmaceuticals-17-00479-f002:**
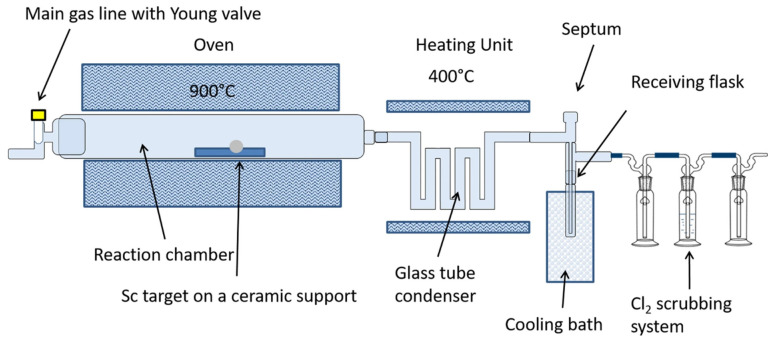
Schematic representation of the separation of ^45^Ti from Sc using a thermochromatographic separation. (Figure used from Giesen et al. [35], licensed under CC by 4.0, https://creativecommons.org/licenses/by/4.0/ (accessed on 24 March 2024)).

**Figure 3 pharmaceuticals-17-00479-f003:**
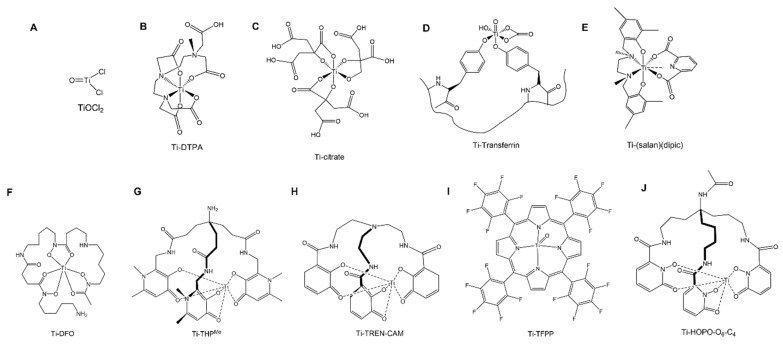
Schematic diagram of the titanium complexed compounds reported in the literature. (**A**) titanium oxychloride (TiOCl_2_), (**B**) titanium-diehtylenetriamine pentaacetate (Ti-DTPA), (**C**) titanium-citrate, (**D**) titanium-transferrin, (**E**) (salan)Ti(dipic), (**F**) titanium-desferrioxamine (Ti-DFO), (**G**) titanium-tris-3,4-hdroxypyridinone (Ti-THP), (**H**) Ti-TREN-CAM, (**I**) Ti-TFPP, (**J**) titanium-tris-1,2-hydroxypyridionone (Ti-HOPO-O_6_-C4).

**Figure 4 pharmaceuticals-17-00479-f004:**
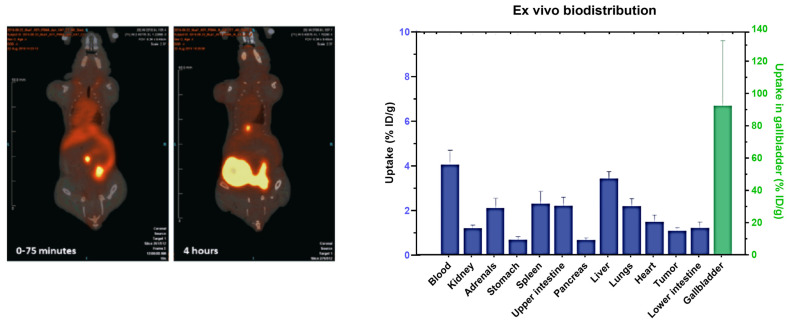
PET/CT images of [^45^Ti]salan-Ti-CA-PSMA in tumor-bearing mice for 0–75 min and 4 h post injection. (Right) Ex vivo biodistribution of tumor-bearing mice at 4 h post injection (figure used from Pederson et al. [36], licensed under CC by 4.0 https://creativecommons.org/licenses/by/4.0/ (accessed on 24 March 2024)).

**Figure 5 pharmaceuticals-17-00479-f005:**
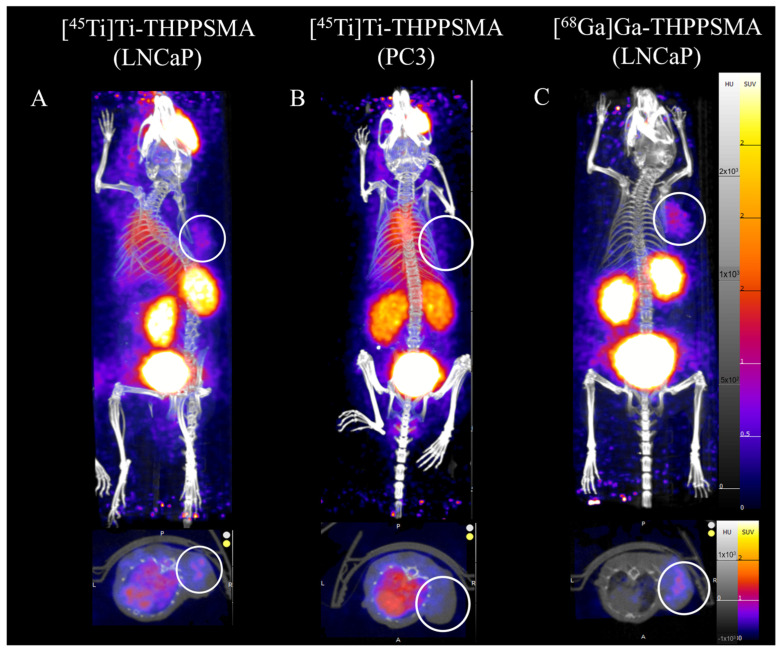
PET/CT images of [^45^Ti]Ti-THP-PSMA for (**A**) LNCaP, (**B**) PC3 tumor-bearing mice and (**C**) [^68^Ga]Ga-THP-PSMA in LNCaP tumor-bearing mice at 1 h post injection (figure adapted with permission from Saini et al. [51]).

**Table 1 pharmaceuticals-17-00479-t001:** Characteristics of radionuclides used for PET imaging.

Isotope	Half-Life (t_1/2_)	Decay	Mean β^+^ Energy (keV)	Reaction	Target Abundance	Cost	References
^45^Ti	184 min	β^+^ (85%)	439	^45^Sc(p,n)^45^Ti	100%	$	[23,24]
^64^Cu	12.7 h	β^+^ (17.8%)	278	^64^Ni(p,n)^64^Cu	1.16%	$$$	[25]
^11^C	20.3 min	β^+^ (100%)	386	^14^N(p,α)^11^C	99.63%		[26,27]
^18^F	110 min	β^+^ (97%)	250	^18^O(p,n)^18^F	0.2%	$$	[28]
^68^Ga	68 min	β^+^ (88%)	830	^68^Ge generator	N/A	$$$	[29]

**Table 2 pharmaceuticals-17-00479-t002:** Production and purification methods for titanium-45.

Resin	Target	Elution	Elution Volume	Radiochemical Yield	Reference
Dowex 1-X8 (100–200 mesh)	Foil	8 M HCl	0.1 mL	30%	[20]
AG 50W x 8 (100–200 mesh)	Foil	6 M HCl	2 mL	75–90%	[41]
AG 50W x 8 (100–200 mesh)	Foil	6 M HCl	6 × 6 mL	92.5%	[23]
AG 50W x 8 (100–200 mesh)	Sc_2_O_3_	6 M HCl	NA	N/A	[42]
HypoGel 200	Foil		3 mL	~30%	[43]
Hydroxamate resin	Foil	Oxalic acid (1 M)	5 mL	N/A	[32]
Liquid–liquid extraction	Foil	guaiacol/anisole		N/A	[34]
Solvent-free	Foil	TiCl_4_	NA	53%	[35]
Hydroxamate resin	Foil	citric acid (1 M)	3 mL	78 ± 8%	[24]
CA-Def *	Foil	HCl (6 M)	0.4 mL	80%	[39]
Hydroxamate resin	Foil	Oxalic acid (0.1 M, pH 3)	2.5 mL	61 ± 8%	[44]

* Deferiprone appended carboxylic acid polymer-functionalized silica resin.

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
