# Peer review of "Titanium-45 (45Ti) Radiochemistry and Applications in Molecular Imaging"

_pharmaceuticals, 2024, doi:10.3390/ph17040479_

Round 1
Reviewer 1 Report
Comments and Suggestions for Authors
The review article is nicely written and precisely describe the current advancement in Titanium 45. Emphasizes the fact that stabilizing ligands might help to achieve clinical use in therapy and diagnostic.
Comments on the Quality of English Language
There are some instances where italics have not been used for example in vivo
Author Response
We thank the reviewer for their time and thoughtful comments. A point-by-point response can be found attached.

Reviewer 2 Report
Comments and Suggestions for Authors
Titanium-45 (45Ti) is a promising radionuclide for the development of positron emission tomography(PET) radiopharmaceuticals. In this manuscript, the authors provide a detailed summary of this radionuclide including production, purification, radiochemistry study, and the application prospects of PET imaging probes. The work sounds interesting, however, the manuscript needs major revision before accepted. My comments are as follows.
1) Page 9, In vitro plasma binding studies should be In vitro plasma binding studies.
2) Page 10, absorption ratio The imaging… should be absorption ratio. The imaging…
3) Page 10, In vivo should be In vivo.
4) Authors should provide chemical structures of some reported [45Ti]Ti-radiolabeled complexes.
5) Authors should add some discussions about the perspective of [45Ti]Ti-labeled radiopharmaceuticals.
6) In particular, there are several format issues in References section.
Author Response

(The authors gave the same response as above.)

Reviewer 3 Report
Comments and Suggestions for Authors
This review gives a very comprehensive aspects of the production of titanium-45 and its application in nuclear medicine.
Very minor suggestion: Use Titanium-45 instead of 45Ti at the start of the sentence. (i.e. Page 2 3rd paragraph)
Author Response

(The authors gave the same response as above.)

Reviewer 4 Report
Comments and Suggestions for Authors
This is a nice comprehensive review, See attached report.

Comments on the Quality of English LanguageThe English is satisfactory with minimal requirements.
Author Response

(The authors gave the same response as above.)

Round 2
Reviewer 2 Report
Comments and Suggestions for Authors
Titanium-45 is a radionuclide with properties suitable for PET imaging. The manuscript by Suzanne E Lapi et al. submitted to Pharmaceuticals described the production, purification, radiochemistry and PET imaging studies of Titanium-45. The merit of this mini-review is that it is concise, but at the same time also rich in information due to the high specificity of the topic. However, some issues should be addressed before publishing this mini-review.
Some revisions are as follows:
1. Page 11, every 15 minutes should be every 15 min.
2. This manuscript appears as a physical mixture of the cited work or information without cohesive presentation and discussion. The authors are encouraged to provide their expert opinions and critical comments.
3. In the conclusion section, if the authors could forecast future directions of Titanium-45 labeled complex for PET imaging, it would be helpful to the readers.
4. In particular, there are several format issues in References section.
Comments on the Quality of English LanguageIt is OK.
